# Mental health of UK Members of Parliament in the House of Commons: a cross-sectional survey

Daniel Poulter,[1] Nicole Votruba,[2] Ioannis Bakolis,[3] Frances Debell,[4] Jayati Das-Munshi,[5] Graham Thornicroft[6]

DP and NV are joint first authors.

For numbered affiliations see end of article.

**Correspondence to**
Nicole Votruba;
nicole.votruba@kcl.ac.uk

## ABSTRACT

**Objectives** The purpose of this study was to assess (1) the overall mental health of Members of Parliament (MPs) and (2) awareness among MPs of the mental health support services available to them in Parliament.

**Design** An anonymous self-completed online cross-sectional survey was conducted in December 2016.

**Setting** 56th UK House of Commons.

**Participants** All 650 members of the 56th UK House of Commons were invited to participate; 146 MPs (23%) completed the survey.

**Outcomes** The General Health Questionnaire-12 was used to assess age- and sex-standardised prevalence of probable common mental disorders (CMD). Results were compared with a nationally representative survey, the Health Survey for England (HSE) 2014. Core demographic questions, MPs' awareness of available mental health services, their willingness to discuss mental health issues with party Whips and fellow MPs and the effects of employment outside Parliament were assessed.

**Results** Comparison of MP respondents with HSE comparator groups found that MPs have higher rates of mental health problems (age- and sex-standardised prevalence of probable CMD in 49 surveyed MPs 34% (95% CI 27% to 42%) versus 17% (95% CI 13% to 21%) in the high-income comparison group). Survey respondents were younger, more likely to be female and more educated compared with all MPs. 77% of MPs (n=112) did not know how to access in-house mental health support. 52% (n=76) would not discuss their mental health with party Whips or other MPs (48%; n=70).

**Conclusions** MPs in the study sample had higher rates of mental health problems than rates seen in the whole English population or comparable occupational groups. Most surveyed MPs are unaware of mental health support services or how to access them. Our findings represent a relatively small sample of MPs. There is a need for MPs to have better awareness of, and access to, mental health support.

## INTRODUCTION

There is a public fascination with understanding the psyches of politicians and decision-makers, from ancient times to the present day, and a long history of public debate about the mental health of politicians, including discussion of the potential

---

**Strengths and limitations of this study**

► This is a unique study where the mental health of MPs has been assessed using structured validated scales for the first time.
► This study is also the first evaluation of MPs' awareness of the mental health support available to them from the Parliamentary Health and Well-being Service and how to access this service.
► This study also assessed for the first time the willingness of MPs to discuss any mental health issues with party Whips or with fellow MPs.
► The survey had a relatively low response rate which may be related to the stigma associated with mental illness and to the nature of an MP's role, which is associated with a stressful work schedule and life in the public eye.

---

psychiatric diagnoses of notable individuals active in political life.[1-9] Research studies have considered some related questions, such as the harassment and stalking of politicians.[10-13] Studies have also examined media and public reactions to politicians' actual or perceived mental health problems.[14-17] Yet little has been published on the actual mental health or mental illness of politicians. Some evidence of politicians disclosing personal mental health problems has been published—for example, during the passage of the UK Mental Health (Discrimination) Act in 2013, which removed discriminatory provisions permitting disqualification of Members of Parliament (MPs) with mental health problems under certain circumstances.[18]

A scoping literature search in January 2017 was conducted to understand what is known about politicians' mental health, and in particular the prevalence of common mental disorders (CMDs) in this group. The papers identified were largely limited to politicians in the UK, USA and Australasia. There remains a dearth of evidence on the prevalence of CMDs in politicians and how this compares with general population rates.

To date, no quantitative ethically approved surveys have been conducted of MPs in the UK Parliament to assess their mental health and to assess their awareness of the available support and treatment services.

Several factors in the UK political system may adversely influence MPs and their mental health. The UK Parliament permits MPs to hold employment outside Parliament in addition to their roles as elected representatives. Further, in the UK Parliament, 'Whips' are appointed officials in each political party who are charged with organising their party's parliamentary business and ensuring party discipline among MPs. In addition, a confidential in-house service is provided within Parliament for MPs and peers, called the Parliamentary Health and Well-being Service, to support their occupational health and well-being.

In this context, the aims of the UK Parliamentary Mental Health (UKPMH) study are to: (1) assess the overall mental health of MPs by drawing comparisons with a nationally representative survey in England and with comparator sociodemographic and occupational groups within the survey; and (2) assess awareness among MPs of the mental health support services available to them.

The principal research question was: What is the prevalence of CMDs among MPs? The secondary questions addressed were: How far are MPs aware of mental health services that can assist them with mental health problems? Are MPs willing to discuss their mental health with party Whips or other MPs? This study tested the primary hypothesis that the occurrence of CMDs is higher among MPs compared with the general population and compared with specific sociodemographic, professional and occupational comparator groups.

## METHODS
### Study design and participants
We conducted an anonymised online self-completed survey at the House of Commons in December 2016. The inclusion criteria for participation were: membership of the 56th UK Parliament, House of Commons and providing written informed consent. We followed the STROBE guidelines for observational studies for the reporting of this cross-sectional study.[19] No age limits were defined, except that to be elected to Parliament one must be aged over 18 years. Participants were sent via email an invitation letter to participate. Initially, in November 2016, a letter was sent to all 650 members of the House of Commons to make them aware of the study. In early December, a letter including a web link to an online survey with an individual access code was sent out to all MPs via internal post and via email. The survey took place between 5 and 31 December 2016. Repeated efforts were taken to promote participation and maximise response rates in the survey. The study information sheet (explaining the purpose of the study) and instructions for the online questionnaire, as well as two reminder emails, were sent out with clear descriptions of encrypted data collection and protection measures to ensure anonymity.

### Ethics and data protection
At all times throughout the study preparation, conduct and analysis, particular consideration and care has been given to the specific sensitive study context and to the potential vulnerability of participants—namely, the risk of sensationalised coverage should any individual be identifiable. Ethics approval for the study was obtained in September 2016 from King's College London Ethics Committee (reference number: HR-16/17–3118). Efforts were taken to limit distress and secure confidentiality for the participants. To ensure full confidentiality, no personal identifiers were collected and identifiers were removed if provided. All participants were provided with contact information for the Parliamentary Health and Well-being Service in the introductory letter and via the online survey in case any participants were experiencing distress at the time of the survey.

### Health Survey for England comparator groups
Data for the comparator groups were elicited from the Health Survey for England (HSE) 2014. The HSE is an annual survey which uses a multi-stage stratified design to sample a nationally representative random cross-section of the population of England each year. Participants are visited by an interviewer who collects demographic and socioeconomic data and information on health and health-related behaviours. A detailed description of the HSE has been reported elsewhere.[20] From the HSE we identified four comparison groups: total population of England in the HSE England population (EN); corporate managers in England (CM); all managers in England (AM); and those in high-income groups in England (HIG). The socioeconomic groups derive from a standardised questionnaire asked in the HSE to all survey respondents.

### Measures of mental health
The General Health Questionnaire (GHQ-12) was used to assess the mental health of respondents in the UKPMH sample and the HSE 2014. The self-completed 12-item GHQ-12 is one of the most extensively used screening instruments for CMDs, measured by a 4-point Likert scale (ranging from 'less than usual' to 'much more than usual') across 12 items.[20 21]

Scoring of the GHQ-12 for the present study was done in the original bimodal method as developed by Goldberg.[22] Specifically, each symptom was scored either 0 if 'not at all present' or present 'no more than usual', or 1 for symptoms that were present 'rather more than usual' or 'much more than usual'. The scoring method allowed for total scores to range from 0 to 12. No formal threshold exists for identifying probable mental ill health, with optimal values likely to be specific to the population under study. However, in line with the previous HSE survey, MPs' total scores are grouped according to three categories:

0 (indicating no evidence of probable mental ill health), 1–3 (indicating less than optimal mental health), and ≥4 (indicating probable psychological disturbance or mental ill health).[20 21]

The GHQ-12 has been extensively validated across international settings for screening and detection of CMDs.[23] In previous work, with a cut-off point of ≥4, the total score of the GHQ-12 was found in a UK setting to have a sensitivity of 84.6% and specificity of 89.3% when assessed against the International Classification of Mental Disorders (ICD-10) and the Diagnostic Statistical Manuals-IV (DSM-IV) diagnoses derived from the Composite International Diagnostic Interview (CIDI-PC) for the CMDs (including depression, dysthymia, generalised anxiety disorder, panic disorder and other related conditions).[23]

A technical error in the administration of the questionnaire caused a lack of indication for respondents of the fourth option (much more/much less than usual) on GHQ-12 items 8, 9, 10, 11, 12. However, this has no impact on the total scores of GHQ-12 for each participant as the third and fourth options are grouped together in the bimodal scoring.

In the question on awareness of the Parliamentary Health and Well-being Service, a technical error in the administration of the questionnaire caused four options (no/unsure/unaware/yes) to be offered rather than binary yes and no options. The three options (no/unsure/unaware) were combined to represent 'no awareness'.

### Covariates

Core demographic questions were obtained from the UKPMH study sample: age (categorised into five groups: 21–30; 31–40; 41–50; 51–60; 61–70, >70 years), sex (female or male) and educational status (GCSE/O level, A Level, Vocational Qualifications, Undergraduate Degree, Postgraduate Degree, Doctorate), as well as years serving as an MP. MPs were also asked if they were aware of the mental health services available to them, as well as their willingness to discuss their mental health with their Whips and other MPs (see full list of questions in the online supplementary file). Ethnicity was not assessed. Due to the low number of MPs from a minority ethnic background in the 56th House of Commons (n=41), this avoided any concern about the identification of participants, which may have further limited the response rate.

### Statistical analyses

All statistical analyses were performed using STATA 14.1. Within the UKPMH sample, descriptive analysis was undertaken first to determine the distribution of each item of the GHQ-12 and of sociodemographic characteristics, awareness of mental health services and willingness to discuss mental health issues with party Whips or with fellow MPs.

The UKPMH sample is subject to 'unit non-response' as 22.4% of all MPs completed the survey. To address this issue, we employed inverse probability weighting[24]

in the analysis, where weights are used to rebalance the set of complete cases within the MP sample to make it representative of the whole English population; we used the weighted sample of the HSE 2014. Age- and sex-standardised proportion estimates were calculated for each item of the GHQ-12 and for the presence of probable mental ill health. We compared each item of the GHQ-12 and the three combined categories derived from the total score of the GHQ-12 that indicate the presence of probable mental ill health of the MP sample with a range of sociodemographic groups (EN, CM, AM and HIG in England) derived from HSE 2014. As sensitivity analyses, age- and sex-standardised proportion estimates were calculated separately for men and women.

Non-parametric tests ($\chi^2$) and parametric tests (t-test for unequal sample sizes) were employed to explore potential differences in the proportion estimates between UKPMH and HSE 2014 samples.

Cross-sectional associations of whether an MP had additional employment outside Parliament with each different item of the GHQ-12 and with the three combined categories (indicating no evidence of probable mental ill health, less than optimal mental health, probable psychological disturbance or mental ill health) were explored with the use of ordinal logistic regression models. The results, expressed as increased risk (OR and corresponding 95% CIs) of being in the highest category of each item of the GHQ-12 for those MPs with a work role outside Parliament, were compared with those without such an external role.

In addition, linear regression models were used to explore the mean difference in the GHQ-12 total scores for those MPs who had additional employment outside Parliament and for those who did not. All models were adjusted for the following potential confounders identified a priori: age, sex and educational status. Age- and sex-standardised inverse probability weights were employed for all linear and ordinal regression models.

### Patient and public involvement

Daniel Poulter, MP, was involved at all stages of the study and is co-author of the paper. Other parliamentarians and staff of the Parliamentary Health and Well-being Service were consulted at the planning and design stages, as well as at the interpretation of the findings and dissemination stages of the study.

### RESULTS

Questionnaires were returned by 146 respondents (22.4%) of the 650 MPs. Median time to complete the survey was 4 min (IQR 3–5). Most respondents were male (63%), with an undergraduate (44%) or postgraduate degree (36%) or doctorate (2%). Most were aged between 41 and 60 years (66%) and most did not work outside Parliament (81%) (see table 1).

### Mental health of MPs and the HSE 2014 comparator groups

Table 2 presents weighted proportion estimates and corresponding 95% CIs of the UKPMH sample and the

**Table 1** Demographic characteristics of participants in the UK Parliamentary Mental Health study

| | MP sample (n=146) | Total Health Survey for England sample (n=7871) |
| --- | --- | --- |
| | n (%) | n (%) |
| Age <40 years | 27 (18) | 4014 (51) |
| Women | 54 (37) | 4385 (55) |
| Higher education degree | 119 (82) | 888 (11.3) |
| Knowledge on how to access mental health support | 65 (45) | N/A |
| Unaware of Parliamentary Well-being Service | 112 (77) | N/A |
| Willing to discuss mental health problems with Whips | 70 (48) | N/A |
| Willing to discuss mental health problems with other MPs | 76 (52) | N/A |
| Presence of CMD (according to ≥4 cut-off point on GHQ-12 total score) | 49 (34) | 2902 (26) |

CMD, common mental disorder.

four different predetermined HSE 2014 occupational and sociodemographic comparator groups (EN, CM, AM, HIG). For each item of the GHQ-12, the UKPMH sample had a higher weighted proportion of participants who had lower levels of concentration, were losing sleep because of worry, were feeling less useful, were less capable of making decisions and were feeling under constant strain compared with the four HSE 2014 occupational and sociodemographic comparison groups (p<0.001, $\chi^2$ test).

In addition, a higher weighted proportion of MPs could not overcome difficulties, were less able to enjoy normal day-to-day activities, were less able to face up to their problems, reported losing confidence in themselves or feeling unhappy and depressed and considered themselves to be a worthless person (p<0.001, $\chi^2$ test). Compared with the HSE 2014 predetermined occupational and sociodemographic comparator groups, a higher weighted proportion of MPs also reported being less able to feel reasonably happy (p<0.001, $\chi^2$ test).

When we compared the weighted proportions of the three combined categories derived for the GHQ-12 total score that indicate the presence of probable mental ill health between the UKPMH and HSE 2014 samples we found that a higher proportion of MPs had probable mental ill health (weighted proportion 34%; 95% CI 27% to 42%) compared with EN (weighted proportion 26%; 95% CI 25% to 27%), CM (weighted proportion 22%; 95% CI 18% to 26%), AM (weighted proportion

23%; 95% CI 20% to 27%) and HIG (weighted proportion 17%; 95% CI 13% to 21%) (p<0.001, $\chi^2$ test) (see table 2 and figure 1). In addition, female MPs had higher rates of probable mental ill health (weighted proportion 41%; 95% CI 27% to 56%) than male MPs (weighted proportion 30%; 95% CI 21% to 41%) (see online supplementary table S1 and S2).

### Characteristics of respondents in comparison with all MPs

Compared with all 650 MPs, those who participated were younger (18% (n=27) < 40 years old vs 16% of total MP population), more likely to be female (37% (n=54) vs 30% of total MP population) and more educated (81% (n=119) of the UKPMH sample had a university degree vs 76% of total MP population).

### Awareness of mental health support services

Most MPs were not aware of the mental health services provided by the Parliamentary Health and Well-being Service within Parliament. Most MPs (55%) did not know how to access any mental health support at Parliament (see figure 2). When asked whether they felt the Parliamentary Health and Well-being Service currently offered sufficient support, a large majority of MPs (77%) were unaware of what options are currently offered by the service and only 23% were aware that support was sufficiently available (see figure 3).

### Willingness to disclose poor mental health

Most MPs who took part in this survey were not willing to discuss mental health problems with their party Whips (52%) and only a small majority of MPs would feel able to talk with other MPs about their mental health (52%) (see figures 4 and 5). After adjusting for age, sex and educational status, we found evidence that MPs who were willing to discuss their mental health with their party Whips or fellow MPs had a reduced risk of CMDs (willing to discuss with Whips: adjusted OR 0.32, 95% CI 0.16 to 0.31, discuss with fellow MPs: adjusted OR 0.57, 95% CI 0.30 to 0.99).

### Additional employment outside Parliament

We found no evidence of an association between having additional employment outside Parliament and the individual GHQ-12 items or an increased total GHQ score indicating poor mental health (see online supplementary table S3).

## DISCUSSION
### Principal findings

The main findings of this study were: (1) strong evidence to indicate that a higher proportion of MPs had poor mental health than among the general population and than among the defined occupational and sociodemographic comparator groups (EN, CM, AM, HIG). The primary study hypothesis was therefore confirmed. (2) Most MPs were not aware of Parliamentary mental health and support services. (3) Most MPs were not willing to discuss their mental health with party Whips, and only a

**Table 2** Descriptive characteristics of the 12-item GHQ (GHQ-12) and the four different predetermined HSE 2014 occupational and sociodemographic comparator groups (EN, CM, AM, HIG)

| | n | WP 95% CI | n | WP 95% CI | n | WP 95% CI | n | WP 95% CI | n | WP 95% CI |
|---|---|---|---|---|---|---|---|---|---|---|
| | **MP** | | **EN** | | **CM** | | **AM** | | **HIG** | |
| **Item 1: Have you recently been able to concentrate on whatever you're doing?** | | | | | | | | | | |
| Better than usual | 5 | 0.03 0.01 to 0.07 | 223 | 0.035 0.03 to 0.04 | 15 | 0.03 0.02 to 0.05 | 24 | 0.03 0.02 to 0.05 | 10 | 0.03 0.01 to 0.05 |
| Same as usual | 93 | 0.66 0.57 to 0.74 | 6073 | 0.85 0.84 to 0.86 | 394 | 0.88 0.84 to 0.91 | 602 | 0.88 0.85 to 0.91 | 371 | 0.9 0.87 to 0.93 |
| Less than usual | 40 | 0.26 0.19 to 0.34 | 771 | 0.1 0.10 to 0.11 | 38 | 0.08 0.06 to 0.11 | 53 | 0.08 0.06 to 0.10 | 29 | 0.07 0.05 to 0.10 |
| Much less than usual | 8 | 0.05 0.02 to 0.11 | 103 | 0.01 0.01 to 0.02 | 2 | 0.01 0.00 to 0.04 | 3 | 0.01 0.00 to 0.03 | 1 | 0.005 0.00 to 0.01 |
| **Item 2: Have you recently lost much sleep over worry?** | | | | | | | | | | |
| Not at all | 24 | 0.18 0.12 to 0.26 | 2334 | 0.33 0.32 to 0.34 | 146 | 0.33 0.28 to 0.38 | 226 | 0.33 0.29 to 0.37 | 130 | 0.3 0.26 to 0.35 |
| No more than usual | 66 | 0.47 0.38 to 0.56 | 3573 | 0.5 0.49 to 0.51 | 246 | 0.54 0.49 to 0.59 | 370 | 0.55 0.50 to 0.59 | 220 | 0.56 0.51 to 0.61 |
| Rather more than usual | 38 | 0.26 0.19 to 0.34 | 1035 | 0.14 0.13 to 0.15 | 51 | 0.11 0.08 to 0.14 | 76 | 0.11 0.09 to 0.14 | 55 | 0.13 0.10 to 0.16 |
| Much more than usual | 18 | 0.1 0.06 to 0.16 | 240 | 0.03 0.02 to 0.04 | 7 | 0.02 0.01 to 0.03 | 11 | 0.02 0.01 to 0.03 | 6 | 0.01 0.00 to 0.03 |
| **Item 3: Have you recently felt you were playing a useful part in things?** | | | | | | | | | | |
| More so than usual | 27 | 0.19 0.13 to 0.27 | 676 | 0.10 0.09 to 0.11 | 58 | 0.16 0.12 to 0.21 | 83 | 0.14 0.11 to 0.18 | 39 | 0.10 0.07 to 0.13 |
| Same as usual | 67 | 0.46 0.38 to 0.55 | 5696 | 0.8 0.79 to 0.81 | 362 | 0.77 0.72 to 0.81 | 548 | 0.78 0.74 to 0.81 | 339 | 0.82 0.77 to 0.85 |
| Less useful than usual | 43 | 0.3 0.22 to 0.39 | 625 | 0.08 0.07 to 0.09 | 26 | 0.07 0.05 to 0.10 | 47 | 0.08 0.06 to 0.10 | 30 | 0.08 0.05 to 0.12 |
| Much less useful | 9 | 0.05 0.02 to 0.11 | 157 | 0.02 0.01 to 0.03 | 3 | 0.005 0.00 to 0.02 | 4 | 0.005 0.00 to 0.02 | 3 | 0.01 0.00 to 0.02 |
| **Item 4: Have you recently felt capable of making decisions about things?** | | | | | | | | | | |
| More so than usual | 9 | 0.06 0.03 to 0.11 | 509 | 0.08 0.07 to 0.09 | 29 | 0.07 0.05 to 0.11 | 42 | 0.07 0.05 to 0.09 | 28 | 0.07 0.05 to 0.10 |
| Same as usual | 118 | 0.84 0.77 to 0.89 | 6162 | 0.85 0.84 to 0.86 | 403 | 0.88 0.84 to 0.91 | 613 | 0.89 0.86 to 0.91 | 367 | 0.89 0.85 to 0.92 |
| Less so than usual | 17 | 0.09 0.05 to 0.15 | 444 | 0.066 0.06 to 0.08 | 17 | 0.04 0.02 to 0.07 | 27 | 0.04 0.03 to 0.06 | 16 | 0.04 0.02 to 0.07 |
| Much less capable | 2 | 0.01 0.00 to 0.05 | 66 | 0.01 0.01 to 0.01 | 1 | 0 0.00 to 0.02 | 1 | 0 0.00 to 0.01 | 0 | NA |
| **Item 5: Have you felt under constant strain recently?** | | | | | | | | | | |
| Not at all | 9 | 0.07 0.03 to 0.13 | 1778 | 0.25 0.24 to 0.27 | 130 | 0.28 0.24 to 0.33 | 194 | 0.28 0.24 to 0.31 | 94 | 0.22 0.18 to 0.27 |
| No more than usual | 60 | 0.41 0.33 to 0.50 | 3974 | 0.56 0.54 to 0.57 | 243 | 0.54 0.49 to 0.59 | 374 | 0.55 0.51 to 0.59 | 236 | 0.57 0.51 to 0.62 |
| Rather more than usual | 53 | 0.38 0.30 to 0.47 | 1192 | 0.16 0.15 to 0.17 | 69 | 0.17 0.13 to 0.21 | 102 | 0.16 0.13 to 0.20 | 75 | 0.19 0.15 to 0.24 |
| Much more than usual | 24 | 0.14 0.09 to 0.21 | 225 | 0.03 0.02 to 0.03 | 7 | 0.02 0.01 to 0.03 | 12 | 0.02 0.01 to 0.03 | 6 | 0.02 0.01 to 0.04 |
| **Item 6: Have you recently felt you couldn't overcome your difficulties?** | | | | | | | | | | |
| Not at all | 41 | 0.29 0.21 to 0.37 | 2659 | 0.38 0.37 to 0.39 | 183 | 0.4 0.35 to 0.45 | 278 | 0.4 0.36 to 0.44 | 156 | 0.36 0.31 to 0.41 |

Continued

**Table 2** Continued

| | n | WP 95% CI | n | WP 95% CI | n | WP 95% CI | n | WP 95% CI | n | WP 95% CI |
|---|---|---|---|---|---|---|---|---|---|---|
| | MP | | EN | | CM | | AM | | HIG | |
| No more than usual | 76 | 0.52 0.44 to 0.61 | 3762 | 0.52 0.51 to 0.53 | 234 | 0.53 0.47 to 0.58 | 352 | 0.52 0.48 to 0.56 | 229 | 0.57 0.52 to 0.62 |
| Rather more than usual | 24 | 0.16 0.10 to 0.23 | 602 | 0.08 0.08 to 0.09 | 31 | 0.07 0.05 to 0.10 | 48 | 0.07 0.05 to 0.09 | 23 | 0.06 0.04 to 0.09 |
| Much more than usual | 5 | 0.03 0.01 to 0.08 | 143 | 0.02 0.02 to 0.02 | 2 | 0.01 0.00 to 0.03 | 5 | 0.01 0.00 to 0.02 | 2 | 0 0.00 to 0.02 |
| **Item 7: Have you recently been able to enjoy your normal day-to-day activities?** | | | | | | | | | | |
| More so than usual | 6 | 0.03 0.01 to 0.06 | 376 | 0.06 0.05 to 0.07 | 35 | 0.11 0.07 to 0.16 | 47 | 0.09 0.06 to 0.13 | 23 | 0.05 0.04 to 0.08 |
| Same as usual | 88 | 0.61 0.52 to 0.69 | 5649 | 0.79 0.78 to 0.80 | 358 | 0.76 0.71 to 0.81 | 544 | 0.77 0.73 to 0.81 | 344 | 0.83 0.79 to 0.87 |
| Less so than usual | 36 | 0.27 0.19 to 0.36 | 924 | 0.12 0.12 to 0.13 | 47 | 0.11 0.08 to 0.14 | 78 | 0.12 0.09 to 0.15 | 40 | 0.11 0.08 to 0.15 |
| Much less than usual | 16 | 0.10 0.06 to 0.16 | 225 | 0.025 0.02 to 0.03 | 9 | 0.02 0.01 to 0.04 | 14 | 0.02 0.01 to 0.03 | 4 | 0.01 0.00 to 0.02 |
| **Item 8: Have you recently been able to face up to your problems?** | | | | | | | | | | |
| More so than usual | 9 | 0.07 0.04 to 0.13 | 340 | 0.06 0.05 to 0.07 | 19 | 0.06 0.04 to 0.11 | 30 | 0.06 0.04 to 0.09 | 17 | 0.05 0.03 to 0.08 |
| Same as usual | 118 | 0.80 0.71 to 0.86 | 6157 | 0.87 0.86 to 0.88 | 404 | 0.90 0.85 to 0.93 | 610 | 0.9 0.86 to 0.92 | 372 | 0.91 0.87 to 0.94 |
| Less able than usual | 19 | 0.14 0.08 to 0.21 | 510 | 0.07 0.06 to 0.07 | 15 | 0.03 0.02 to 0.06 | 27 | 0.04 0.03 to 0.06 | 17 | 0.04 0.02 to 0.07 |
| Much less able | NA | NA | 72 | 0.01 0.01 to 0.01 | 1 | 0.01 0.00 to 0.03 | 1 | 0.01 0.00 to 0.02 | 1 | 0.01 0.00 to 0.02 |
| **Item 9: Have you recently been feeling unhappy and depressed?** | | | | | | | | | | |
| Not at all | 43 | 0.3 0.22 to 0.38 | 2846 | 0.4 0.39 to 0.42 | 213 | 0.47 0.42 to 0.52 | 318 | 0.47 0.43 to 0.51 | 168 | 0.39 0.34 to 0.44 |
| No more than usual | 59 | 0.42 0.33 to 0.51 | 3119 | 0.44 0.43 to 0.45 | 178 | 0.42 0.37 to 0.47 | 271 | 0.41 0.37 to 0.46 | 202 | 0.52 0.47 to 0.58 |
| Rather more than usual | 44 | 0.29 0.21 to 0.37 | 911 | 0.13 0.12 to 0.15 | 44 | 0.1 0.08 to 0.14 | 70 | 0.11 0.08 to 0.13 | 34 | 0.08 0.06 to 0.11 |
| Much more than usual | NA | NA | 206 | 0.03 0.01 to 0.04 | 3 | 0.01 0.00 to 0.02 | 7 | 0.01 0.01 to 0.03 | 3 | 0.01 0.00 to 0.02 |
| **Item 10: Have you recently been losing confidence in yourself?** | | | | | | | | | | |
| Not at all | 53 | 0.37 0.29 to 0.46 | 3192 | 0.45 0.44 to 0.47 | 232 | 0.52 0.47 to 0.58 | 349 | 0.52 0.48 to 0.56 | 201 | 0.47 0.42 to 0.53 |
| No more than usual | 65 | 0.45 0.36 to 0.54 | 2979 | 0.42 0.41 to 0.43 | 175 | 0.4 0.35 to 0.45 | 261 | 0.39 0.35 to 0.43 | 174 | 0.44 0.39 to 0.50 |
| Rather more than usual | 28 | 0.18 0.13 to 0.26 | 739 | 0.1 0.10 to 0.11 | 24 | 0.06 0.04 to 0.10 | 46 | 0.08 0.06 to 0.10 | 32 | 0.08 0.06 to 0.12 |
| Much more than usual | NA | NA | 170 | 0.02 0.02 to 0.03 | 5 | 0.01 0.00 to 0.02 | 9 | 0.015 0.01 to 0.02 | NA | NA |
| **Item 11: Have you recently been thinking of yourself as a worthless person?** | | | | | | | | | | |
| Not at all | 86 | 0.58 0.49 to 0.66 | 4689 | 0.66 0.65 to 0.68 | 323 | 0.73 0.68 to 0.77 | 480 | 0.72 0.68 to 0.75 | 285 | 0.69 0.64 to 0.74 |
| No more than usual | 44 | 0.31 0.24 to 0.40 | 1879 | 0.26 0.25 to 0.27 | 95 | 0.22 0.18 to 0.26 | 154 | 0.23 0.20 to 0.27 | 107 | 0.27 0.23 to 0.32 |
| Rather more than usual | 16 | 0.11 0.06 to 0.18 | 378 | 0.05 0.05 to 0.06 | 16 | 0.05 0.03 to 0.08 | 26 | 0.05 0.03 to 0.07 | 13 | 0.03 0.02 to 0.06 |

**Table 2** Continued

| | n | 95% CI | n | 95% CI | n | 95% CI | n | 95% CI | n | 95% CI |
|---|---|---|---|---|---|---|---|---|---|---|
| | | WP | | WP | | WP | | WP | | WP |
| | MP | | EN | | CM | | AM | | HIG | |
| Much more than usual | NA | NA | 133 | 0.02 0.02 to 0.02 | 3 | 0.01 0.00 to 0.02 | 6 | 0.01 0.00 to 0.02 | 2 | 0.01 0.00 to 0.02 |
| **Item 12: Have you recently been feeling reasonably happy, all things considered?** | | | | | | | | | | |
| More so than usual | 16 | 0.09 0.05 to 0.15 | 698 | 0.11 0.10 to 0.11 | 45 | 0.13 0.09 to 0.18 | 66 | 0.12 0.09 to 0.15 | 39 | 0.11 0.08 to 0.14 |
| About same as usual | 96 | 0.67 0.59 to 0.75 | 5633 | 0.79 0.78 to 0.80 | 364 | 0.8 0.75 to 0.85 | 553 | 0.81 0.77 to 0.84 | 346 | 0.84 0.80 to 0.88 |
| Less so than usual | 34 | 0.24 0.17 to 0.32 | 611 | 0.08 0.08 to 0.09 | 25 | 0.05 0.04 to 0.08 | 42 | 0.06 0.04 to 0.08 | 20 | 0.05 0.03 to 0.08 |
| Much less than usual | NA | NA | 137 | 0.02 0.02 to 0.02 | 4 | 0.01 0.00 to 0.03 | 7 | 0.01 0.01 to 0.03 | 2 | 0 0.00 to 0.02 |
| **Presence of probable mental ill health** | | | | | | | | | | |
| No evidence of probable mental ill health | 35 | 0.25 0.18 to 0.34 | 4256 | 0.53 0.52 to 0.55 | 290 | 0.58 0.53 to 0.62 | 446 | 0.58 0.54 to 0.62 | 254 | 0.56 0.51 to 0.61 |
| Less than optimal mental ill health | 62 | 0.40 0.32 to 0.49 | 1620 | 0.2 0.19 to 0.21 | 97 | 0.2 0.17 to 0.25 | 140 | 0.19 0.16 to 0.22 | 117 | 0.27 0.23 to 0.32 |
| Probable mental ill health | 49 | 0.34 0.27 to 0.43 | 2141 | 0.26 0.25 to 0.27 | 108 | 0.22 0.18 to 0.26 | 170 | 0.23 0.20 to 0.27 | 74 | 0.17 0.13 to 0.21 |

Weighted proportion (WP) with corresponding 95% CI.
AM, All managers (HSE 2014); CM, Corporate managers (HSE 2014); EN, English population (HSE 2014); HIG, high-income group (HSE 2014); MP, Member of Parliament sample.

small majority would be happy to discuss mental health issues with other MPs. (4) Having employment outside Parliament, in addition to the role of MP, is not linked with increased risk for mental ill health.

The Parliamentary Health and Well-being Service is the occupational health service provided since 2013 inside the House of Commons. It aims to support all staff and MPs in developing a healthy and safe working environment, and encourages MPs to adopt better attitudes and behaviour towards their own physical health and mental health.[25] Despite the service being in place for almost

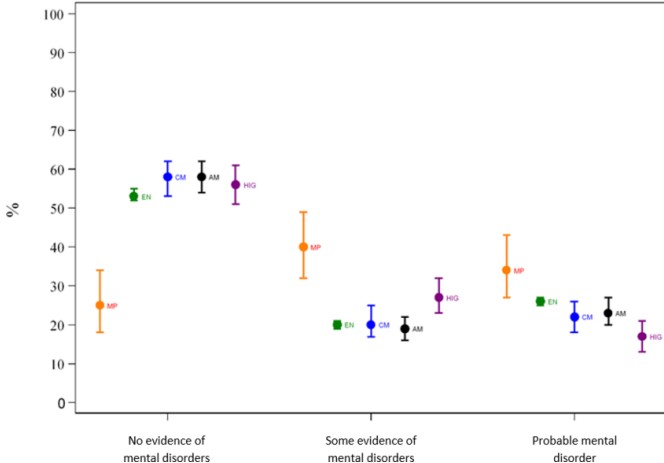

**Figure 1** Age- and sex-standardised prevalence estimates and 95% CIs of UKPMH and of specific population groups of HSE 2014 for the three different categories of common mental disorders. MP, Member of Parliament sample; EN, English population (HSE 2014); CM, Corporate managers (HSE 2014); AM, All managers (HSE 2014); HIG, High-income group (HSE 2014).

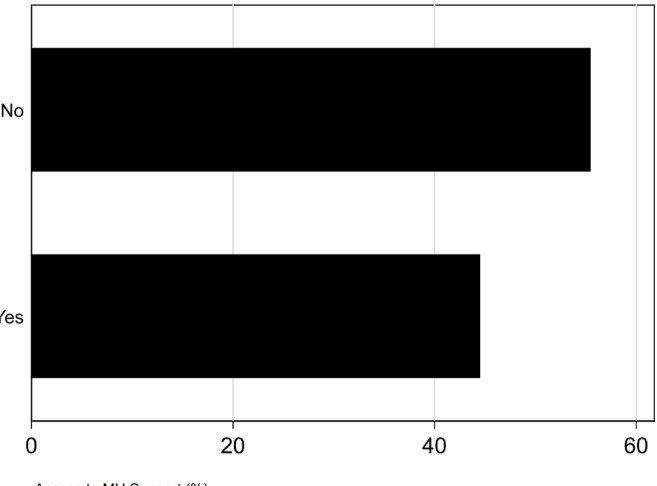

**Figure 2** Access to the mental health (MH) support of the Parliamentary Health and Well-being Service (all p values <0.001).

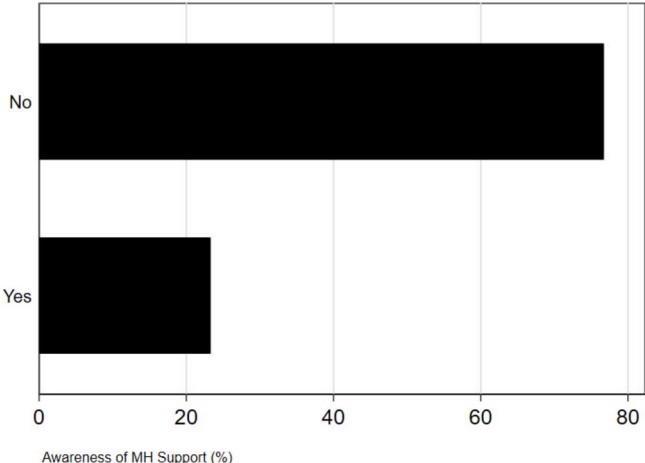

**Figure 3** Awareness of the mental health (MH) support of the Parliamentary Health and Well-being Service (all p values <0.001).

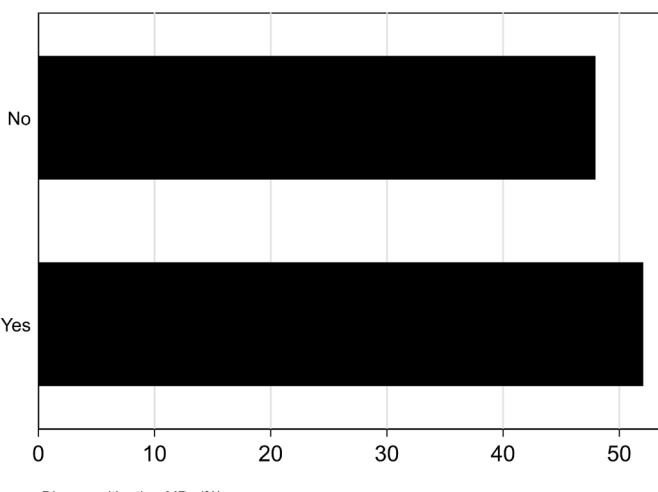

**Figure 5** Willingness to talk to other MPs (all p values <0.001).

4 years, the Parliamentary Health and Well-being Service reported low numbers of MPs requesting support. This study confirms the reluctance to seek help in finding that a majority of MPs are unaware of the service or how to access it. Reasons for this might be insufficient advertising of the support options offered and location of the services, as well as anticipated stigma and discrimination among MPs.[26]

### Strengths and weaknesses of the study

The study has several limitations and potential biases. First, the response rate was relatively low (22.4%). Given the intense workloads of MPs, this may have been partly due to the additional workload of completing the survey, even though the median time to complete the survey was only 4 min. Notably, a possible fear of being identified, of stigmatisation and of the potential reputational damage associated with adverse media coverage may have influenced the response rate. We tried to reduce these biases by promoting the survey in Parliament, by sending

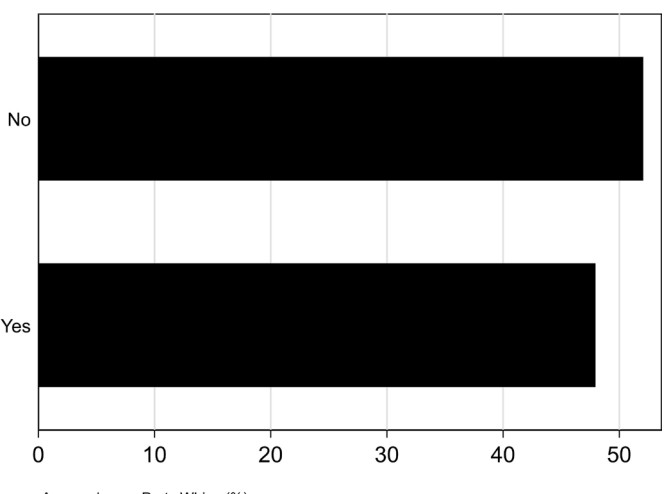

**Figure 4** Willingness to talk to party Whips (all p values <0.001).

several reminders and by stressing the brevity as well as the anonymity of the survey. Generally, MPs are a difficult survey population to engage, which has also been confirmed in a 2008 internal UK Parliament survey to which only 14.5% (94 MPs) responded.[27]

Second, it is also possible that MPs who responded to the online survey may have increased stress or mental ill health and that therefore a greater number of them were willing to complete the survey. A potential self-selection bias may therefore be present in the UKPMH sample. However, there is also a potential risk of under-reporting from people who might be reluctant to take part in the study because they are affected by mental health problems or because of the stigma associated with the topic. Prior experiences of, or fears of stalking and harassment, which might result from their disclosure, may decrease the willingness of MPs to participate in the survey.[28]

Respondents tended to be younger in relation to the age distribution of all MPs (18% of the UKPMH sample vs 16% of the total MP population were aged <40 years), more likely to be female (36% of the UKPMH sample vs 30% of the total MP population were female) and had a university degree (81% of the UKPMH sample vs 76% of the total MP population). We did not assess marital or cohabitation status as this would have increased the risk of identifiability of MPs and may therefore have also adversely affected the response rate.

Third, comparing MPs with other occupational and sociodemographic groups within a population presents challenges. We considered comparing the UKPMH sample with the UK Health and Safety Executive's Labour Force Survey (LFS), which provides annual data on rates of mental disorder by occupation.[29] However, the LFS relies on random household sampling and is poorly suited to extrapolating meaningful data for a relatively small group of 650 UK MPs. Published LFS data lack sufficient granularity to be able to analyse the prevalence of mental disorders at an occupation-specific level which, for politicians, would be 'elected officers and

representatives'.[30] Given the unique features of political careers, including the diverse backgrounds from which politicians may be drawn, specific data relating to these generic occupational groupings are unlikely to be fully helpful in understanding why there is a higher burden of mental ill health. In this sample we found that having employment outside Parliament, and in addition to the role of MP, does not seem to constitute an increased risk for mental ill health. However, we regard this outcome with caution as this study may be underpowered to test for this specific variable, as most participants (81%) did not have employment outside Parliament.

## Comparison of results with earlier studies

When examining UK parliamentary working hours reform, research found high levels of physical and emotional stress as a result of various aspects of political life such as additional work roles, extensive travel and job insecurity.[31] A longitudinal study in new UK MPs highlighted increased levels of stress post-election.[32] In 2008 the UK Parliament also conducted its own informal survey regarding experience and perceptions of mental illness, which concluded that one in five MPs had personal experience of a mental health problem and one in three felt stigma was a barrier to openness about mental health, yet no data on CMD were collected.[27] Given that work characteristics promoting stress are associated with mental disorders,[33 34] it may be reasonable to assume that rates of CMD would be high in parliamentarians. However, no rigorous assessment has previously been conducted to investigate this issue.

Selected studies have investigated mental health in politicians, and although they have drawn on biographical evidence, their findings are in line with the results of this study. One study rated 46 statesmen and national leaders' biographies for psychopathology and found increased rates of lifetime psychopathology and episodes of mental ill health, with only 15.2% of politicians showing no psychopathology at all.[35] A review of biographical sources looking at mental disorders in US Presidents between 1776 and 1974 found that 18 (49%) Presidents met criteria indicative of psychiatric disorders.[36]

A cross-national study in the UK, Australia, New Zealand and Norway found that a higher proportion of MPs than the general public experience stalking, harassment and intrusive or aggressive behaviours.[28] They found that, in the UK, 81% of MPs had experienced intrusive or aggressive behaviours, 18% had been subject to attack/attempted attack and 53% had been stalked or harassed. These intimidating experiences have a negative impact on MPs' mental health and are likely to reinforce stigma and non-disclosure.[37]

This is the first study of assessment of mental health in members of Parliament of the UK House of Commons using structured validated scales. These findings indicate that MPs are more likely to experience probable mental ill health and symptoms indicative of mental distress compared with the general population, and compared

with similar occupational and professional groups. In addition, most MPs are not aware of mental health support offered by the Parliamentary Health and Well-being Service or are willing to disclose to their Whips or other MPs. This leaves MPs who have experience of mental ill health facing considerable difficulties without knowing how to access help.

## Interpretation of the results

A number of studies have examined media and public reactions to politicians' actual or perceived mental health problems.[14–16] In an ever more hostile media environment, poor mental health can be regarded as a factor limiting politicians in their capacities. Stigma against people with mental disorders is prevalent in all countries and all sectors of society. It was not until 2013 that the UK passed the Mental Health (Discrimination) (No 2) Act 2013, which removed discriminatory provisions permitting MPs with mental health problems to be disqualified under certain circumstances.[18] Subsequent to the Act, there have been more disclosures from politicians about personal mental health problems. However, given that the results of this study showed that only 48% of surveyed MPs felt able to talk to their party Whips and only about half (52%) felt able to talk to another MP about their mental health, stigma and self-stigma about mental health appears to remain a powerful barrier to seeking help and support among Members of the UK House of Commons.

The power of disclosure as a catalyst for overcoming stigma has been demonstrated in 1998 when Kjell Magne Bondevik, then Prime Minister of Norway, spoke publicly about his experience of depression. His disclosure was empathetically received by the media and by the public.[38]

In 2012, during a House of Commons debate on mental health, four MPs disclosed their own mental health experiences. This eventually paved the way to providing MPs with access to mental health services in Westminster. Consequently, the Parliamentary Health and Well-being Service was created in 2013 and operates a mental health referral service as well as providing general medical advice, support and guidance to MPs and other staff working at Parliament. The service is nurse-led and is supported by one occupational health doctor for 3 days each week. It does not offer the more comprehensive health service that is often provided by general practice in the UK. Our findings show poor awareness among MPs of the Parliamentary Health and Well-being Service and how to access it. This may be related to the restricted times that the service operates, or that the service is not located on the main Parliamentary Estate. These findings support the need for increased mental health support for MPs and raising awareness about the Parliamentary Health and Well-being Service. They also support the need for mental health stigma and self-stigma reduction among MPs.

## Implications for future research

This is an initial study into the mental health of MPs, and further work is needed to assess the key issues identified and trends in the mental health of MPs over time. Our findings are only a starting point, but they reveal MPs' mental health problems and the need to properly assess them. A more granular assessment of mental health problems—including rates and consequences of problems related to alcohol and substance use—as well as cognitive impairment would be needed to provide a more in-depth picture. In terms of prevention, a better understanding of the causes of mental health problems and specific risk factors in MPs such as (cyber) bullying, harassment or stalking would be informative, and investigating effective mechanisms and strategies for prevention and increasing resilience. There is a need for better promotion of mental health support, such as the Parliamentary Health and Well-being Service, and for additional information and support for MPs in accessing the full range of mental health care. Due to their working routine and hours, MPs spend a majority of their working time far from the support provided by the NHS services in their own constituencies. In addition to their high-performance work life, this adds to the increased stress on MPs' mental health. It is also why strengthening the Parliamentary Health and Well-being Service could offer a specifically relevant support function. Research is also needed on the mental health of other Parliamentary staff to identify their needs and to evaluate their awareness of, and access to, the Parliamentary Health and Well-being Service and other relevant services.

## CONCLUSION AND POLICY IMPLICATIONS

MPs have a vital role to play in the UK democracy in making and scrutinising the legislation that governs the country as well as in representing the interests of their constituents and the nation. This study has found that the people in these important roles experience significantly higher levels of mental ill health compared with the general population and compared with other senior executive and managerial groups. Most MPs do not feel that they have adequate mental health support, and they lack knowledge of how to access the mental health services that are available to them. Most MPs are not able to discuss their mental health problems with their Whips or other MPs. These findings indicate that better support is required both to prevent mental health problems among MPs and to ensure rapid and effective care when needed, to support MPs in their vital work for the people they serve.

**Author affiliations**
[1]House of Commons, London, UK
[2]Centre for Global Mental Health, Institute of Psychiatry, Psychology and Neuroscience, King's College London, London, UK
[3]Department of Biostatistics and Health Informatics/ Centre for Implementation Science, Institute of Psychiatry, Psychology and Neuroscience, King's College London, London, UK
[4]South London and Maudsley NHS Foundation Trust, London, UK
[5]Department of Psychological Medicine, Institute of Psychiatry, Psychology & Neurosciences, King's College London, Institute of Psychiatry, London, UK
[6]Centre for Global Mental Health, Institute of Psychiatry, Psychology and Neuroscience, King's College London, London, UK

**Correction notice** This article has been corrected since it was published. Joint first authorship statement is added.

**Acknowledgements** We would like to thank all Members of Parliament who took part in this study. In addition, we would like to thank Elaine Bryce (member of Dr Daniel Poulter's Parliamentary office) and the staff of the Parliamentary Health and Wellbeing Service for their support in this study.

**Contributors** DP and GT conceived the original idea for the study, which was then discussed with NV. NV coordinated the study. All authors contributed to the design of the study. NV and FD conducted the literature review. DP and NV collected the data. IB conducted design and analysis of the data. JD supported the design of the data analysis and contributed throughout the design and writing up of the study. NV led the writing of the manuscript, and all authors contributed and critically revised it. All authors have given their approval for the publication of this manuscript and agree to be accountable for all aspects of the work to ensure that the questions related to the accuracy or integrity of any part of the work are appropriately investigated and resolved.

**Funding** The authors have not declared a specific grant for this research from any funding agency in the public, commercial or not-for-profit sectors.

**Competing interests** NV acknowledges funding from the Economic and Social Research Council (ESRC) and National Institute for Health Research (NIHR) Collaboration for Leadership in Applied Health Research and Care South London at King's College London NHS Foundation Trust. GT is supported by the National Institute for Health Research (NIHR) Collaboration for Leadership in Applied Health Research and Care South London at King's College London NHS Foundation Trust. The views expressed are those of the author(s) and not necessarily those of the NHS, the NIHR or the Department of Health. GT acknowledges financial support from the Department of Health via the National Institute for Health Research (NIHR) Biomedical Research Centre and Dementia Unit awarded to South London and Maudsley NHS Foundation Trust in partnership with King's College London and King's College Hospital NHS Foundation Trust. GT is supported by the European Union Seventh Framework Programme (FP7/2007-2013) Emerald project. GT also receives support from the National Institute of Mental Health of the National Institutes of Health under award number R01MH100470 (Cobalt study). GT is also supported by the UK Medical Research Council in relation to the Emilia (MR/S001255/1) and Indigo Partnership (MR/R023697/1) awards. IB is supported by the NIHR Biomedical Research Centre at South London and Maudsley NHS Foundation Trust and by the NIHR Collaboration for Leadership in Applied Health Research. JD has a Clinician Scientist Fellowship funded by the Health Foundation working with the Academy of Medical Sciences. DP is currently MP of the 57th UK Parliament and was member of the 56th UK Parliament

**Patient consent for publication** Not required.

**Ethics approval** Ethics approval for the study was obtained in September 2016 from King's College London Ethics Committee (reference number: HR-16/17-3118).

**Provenance and peer review** Not commissioned; externally peer reviewed.

**Data sharing statement** No additional data available. The Health Survey for England 2014 can be accessed at: https://digital.nhs.uk/data-and-information/publications/statistical/health-survey-for-england/health-survey-for-england-2014. Due to the sensibility of the data, and in order to ensure full anonymity, confidentiality and data protection for the participants, the full survey data cannot be made accessible to the public.

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
