## [Reviewer comments · BMJ Open]

ARTICLE DETAILS

TITLE (PROVISIONAL)	Mental health of UK Members of Parliament in the House of Commons: a cross-sectional survey
AUTHORS	Poulter, Daniel; Votruba, Nicole; Bakolis, Ioannis; Debell, Frances; Das-Munshi, Jayati; Thornicroft, Graham

VERSION 1 - REVIEW

REVIEWER	Jonathan Davidson Duke University Medical Center, Durham, NC, USA Financial disclosure: As author of "Downing Street Blues", I declare the receipt of royalties in the past three years..
REVIEW RETURNED	19-Nov-2018

GENERAL COMMENTS	The authors have addressed an important topic that has received only limited attention in the literature. Clearly the mental health of politicians is one factor of importance in their ability to govern effectively. This report builds on earlier pathographic work by others and represents the first "in vivo" study of parliamentarians while in office, with the use of a psychometrically valid scale, the GHQ-12. Novel elements include the measurement of rates of psychopathology and assessment of knowledge about, and attitudes to, the available counselling services on hand. Strengths and limitations are fully acknowledged by the authors. Suggestions and comments are as follows: Abstract, page 3, line 37. There appears to be a typo, in that the overall rate of mental health problems in the HSE is given as 37% in the text and figures, and not 17%. The sample is predominantly male. Were there any differences between the two groups on the key outcome variables? The 34% rate of current mental health issues is almost certainly lower than the lifetime rates would have been in this sample. It is interesting that the current rate is in line with the published rates in US presidents and also politicians internationally by Felix Post in his 1994 B J Psychiatry article, which could be cited either in the introduction or discussion. Page 13 addresses willingness to disclose poor mental health. One is reminded here of Kjell Bondevik, who could also be mentioned in the discussion as an example of the benefits of public disclosure.
---

	It would be useful to describe a bit more about the PHWS, such as its staffing, and also whether or not other health needs are catered for such as having in house family practice services, where much of the world's psychopathology is dealt with. Implications for future research, page 17. Among the more important priorities to investigate would be a more granular assessment of specific forms of psychopathology, such as rates and/or consequences of alcohol/drug-related problems, of which there have been many instances in the past. For older MPs, one would think of cognitive impairment. Depression and grief have been quite seriously impairing for some politicians. The authors note their low response rate, which is not too surprising. One reason not to be over-alarmed is that their findings are line with those of others, as noted above. Figure 3. What is the difference between being unaware and "No awareness" which is presented separately? Some minor issues - page 15, replace "would've" with "would have", and page 20, ref 2 spelling "Dominique Strauss-Kahn" in upper case.
--	---

REVIEWER	Dr. Antonio Lora Director Department of Mental Health - Azienda Socio-Sanitaria Territoriale di Lecco (Italy)
REVIEW RETURNED	27-Nov-2018

GENERAL COMMENTS	The paper is interesting because it addresses a very sensitive and debated matter, specially in the era of populism. The low level of responses (only one out of four) is probably due to the fear of stigmatization rather than to lack of time (four minutes!), as suggested by the authors.
--

REVIEWER	David James Consultant forensic psychiatrist Theseus LLP, UK and adviser to Fixated Threat Assessment Centre, London, UK
REVIEW RETURNED	23-Dec-2018

GENERAL COMMENTS	This is a well-designed and executed study on a subject of importance. I have only a few questions and comments. 1) The authors have surveyed the English language literature about stress and mental health problems in politicians. It is unclear whether the absence of any comment about the mental health of politicians in other countries might be a consequence of a lack of relevant studies or the result of the authors not examining governmental reports in other languages. 2) The sample concerns the 650 MPs elected into the 57th UK parliament. It seems odd to have included the four Sinn Fein members who did not take up their seats. 3) The 22.5% response rate may have disappointed the authors, but it is respectable in terms of surveys of MPs. One problem which might be worth mentioning is that many MPs do not open their own mail: rather, it is filtered by office staff. Surveys in
---

	consequence may never be seen by a proportion of the MPs to whom they are addressed. 4) Did the authors have any data as to how many of those responding to the survey had ministerial positions of any type? It would have been interesting to know if there were any difference between those with government posts and the ordinary backbencher. This would also have been interesting in terms of response rates. 5) The study limited itself to a narrow and clearly defined topic, and there are clearly questions which would be interesting to include in any further studies. The paper does have sections on Interpretation and Implications for further research. The main focus here is on the need for specific support services for MPs. However, a principal question for further research would be upon why a higher of MPs have mental health problems than members of the general population. The authors quote studies about stress and about the psychological effects of stalking and harassment. Some comment as to exploring why MPs have problems could be included in the later sections of the paper. 6) A minor point about punctuation. There has to be a full-stop before 'however' at line three of page 16.
--	--

VERSION 1 – AUTHOR RESPONSE

Reviewers' Comments to Author:

Reviewer: 1

Reviewer Name: Jonathan Davidson

Institution and Country: Duke University Medical Center, Durham, NC, USA

Please state any competing interests or state 'None declared': Financial disclosure: As author of "Downing Street Blues", I declare the receipt of royalties in the past three years.

The authors have addressed an important topic that has received only limited attention in the literature. Clearly the mental health of politicians is one factor of importance in their ability to govern effectively. This report builds on earlier pathographic work by others and represents the first "in vivo" study of parliamentarians while in office, with the use of a psychometrically valid scale, the GHQ-12. Novel elements include the measurement of rates of psychopathology and assessment of knowledge about, and attitudes to, the available counselling services on hand.

Strengths and limitations are fully acknowledged by the authors.

Suggestions and comments are as follows:

1. Abstract, page 3, line 37. There appears to be a typo, in that the overall rate of mental health problems in the HSE is given as 37% in the text and figures, and not 17%.

We apologise for any confusion, as the wording in the abstract was not very clear. We confirm there is no discrepancy. For surveyed MPs we found 34%. The 17% refers to the HSE HIG high-income comparison group. We have clarified this in the abstract (line 45f).

2. The sample is predominantly male. Were there any differences between the two groups on the key outcome variables?

We have added further commentary and explanation to this. There were 92 males and 54 females in our survey. We have added two tables clarifying the differences on the key outcome variables for males and females in the supplementary appendix (Table S2 and S3).

3. The 34% rate of current mental health issues is almost certainly lower than the lifetime rates would have been in this sample. It is interesting that the current rate is in line with the published rates in US presidents and also politicians internationally by Felix Post in his 1994 B JPsychiatry article, which could be cited either in the introduction or discussion.

We agree and we have added both of these studies to the discussion (line 362-368).

4. Page 13 addresses willingness to disclose poor mental health. One is reminded here of Kjell Bondevik, who could also be mentioned in the discussion as an example of the benefits of public disclosure.

We have added this point to the discussion (lines 390-392).

5. It would be useful to describe a bit more about the PHWS, such as its staffing, and also whether or not other health needs are catered for such as having in house family practice services, where much of the world's psychopathology is dealt with.

We have added these details to the discussion (lines 393-402).

6. Implications for future research, page 17. Among the more important priorities to investigate would be a more granular assessment of specific forms of psychopathology, such as rates and/or consequences of alcohol/drug-related problems, of which there have been many instances in the past. For older MPs, one would think of cognitive impairment. Depression and grief have been quite seriously impairing for some politicians.

We have added details to the future research in the discussion (lines 410-413).

7. The authors note their low response rate, which is not too surprising. One reason not to be over-alarmed is that their findings are in line with those of others, as noted above.

We have addressed the response rate in more detail in the discussion (lines 314-318). This was also suggested by the reviewer 2.

8. Figure 3. What is the difference between being unaware and "No awareness" which is presented separately?

This was an error on our part. In the question on awareness of the Parliamentary Health and Wellbeing Service, a technical error in the administration of the questionnaire has caused 4 options (no/ unsure/ unaware/ yes) to be offered rather than a binary yes and no option. We have combined the three options (no/ unsure/ unaware) to represent "no awareness".

We have updated the numbers, and clarified this in abstract (line 47) methods (lines 177-180), and results (Table 1, lines 232, 276).

9. Some minor issues - page 15, replace "would've" with "would have", and page 20, ref 2 spelling "Dominique Strauss-Kahn" in upper case.

We have corrected both typos (lines 334; references)

Reviewer: 2

Reviewer Name: Dr. Antonio Lora

Institution and Country: Director Department of Mental Health - Azienda Socio-Sanitaria

Territoriale di Lecco (Italy)

Please state any competing interests or state 'None declared': none declared

The paper is interesting because it addresses a very sensitive and debated matter, specially in the era of populism. The low level of responses (only one out of four) is probably due to the fear of stigmatization rather than to lack of time (four minutes!), as suggested by the authors.

We agree re low level of responses as a result of fear of stigmatisation, and we have stressed this in the discussion (lines 314-318; 385-392). This was also suggested by the reviewer 1.

Reviewer: 3

Reviewer Name: David James

Institution and Country: Consultant forensic psychiatrist, Theseus LLP, UK and adviser to

Fixated Threat Assessment Centre, London, UK

Please state any competing interests or state 'None declared': None declared

This is a well-designed and executed study on a subject of importance. I have only a few questions and comments.

1. The authors have surveyed the English language literature about stress and mental health problems in politicians. It is unclear whether the absence of any comment about the mental health of

politicians in other countries might be a consequence of a lack of relevant studies or the result of the authors not examining governmental reports in other languages.

□ Essentially, the database searches included English language search terms (politician/parliament/senator/legislature/senate) and no relevant reports/papers from non-English speaking countries were identified amongst the results. Non-English results were not excluded from the searches, so theoretically they would have been captured if the abstract or a duplicate article were available in English, although I suppose there may be non-English language results that would not have been found. Therefore, it is probably a combination of a lack of relevant studies, plus the non-availability of some non-English results via the database searches.

□ We have clarified this in the introduction (lines 85-90).

2. The sample concerns the 650 MPs elected into the 57th UK parliament. It seems odd to have included the four Sinn Fein members who did not take up their seats.

□ We have sent out the survey to all 650 MPs, including the four members of Sinn Fein. Even though these members may not take up their seats, they still receive emails, and attend various Parliamentary events. We have therefore strong reason to believe that our (repeated) invitations to participate were seen also by the four members of Sinn Fein. As our survey was anonymous it is impossible to say, but we decided to leave all 650 MPs as invitees in the manuscript and data analysis.

3. The 22.5% response rate may have disappointed the authors, but it is respectable in terms of surveys of MPs. One problem which might be worth mentioning is that many MPs do not open their own mail: rather, it is filtered by office staff. Surveys in consequence may never be seen by a proportion of the MPs to whom they are addressed.

□ We agree with this point however, we have make every effort to reach the MPs directly and are fairly sure they did indeed receive the invitation personally. The invitation and two reminders were sent to their office MP email address and to their private MP email address. Letters were sent to their offices. The survey was announced twice in different committees, such as the 1922 Committee and the PLP.

□ We have stressed this now also in the discussion (line 317).

4. Did the authors have any data as to how many of those responding to the survey had ministerial positions of any type? It would have been interesting to know if there were any difference between those with government posts and the ordinary backbencher. This would also have been interesting in terms of response rates.

□ We agree, however, for reasons of fear of disclosure and stigma, and to maximise response rates, we did not assess such data.

5. The study limited itself to a narrow and clearly defined topic, and there are clearly questions which would be interesting to include in any further studies. The paper does have sections on Interpretation and Implications for further research. The main focus here is on the need for specific support services for MPs. However, a principal question for further research would be upon why a higher of MPs have mental health problems than members of the general population. The authors quote studies about

stress and about the psychological effects of stalking and harassment. Some comment as to exploring why MPs have problems could be included in the later sections of the paper.

We have elaborated on the risks and potential causes of mental health problems for MPs in the discussion (lines 410-421).

6. A minor point about punctuation. There has to be a full-stop before 'however' at line three of page 16.

We have corrected the typo (line 360).

VERSION 2 – REVIEW

REVIEWER	Jonathan Davidson Duke University Medical Center USA Royalty income from Downing Street Blues in past three years
REVIEW RETURNED	18-Mar-2019

GENERAL COMMENTS	Authors have well addressed questions from previous review.
---

REVIEWER	David James Theseus, UK
REVIEW RETURNED	09-Apr-2019

GENERAL COMMENTS	The authors have not responded as I would have hoped to two of my initial comments. I would recommend publication as is, should the authors not wish at this stage to address them. However, I am repeating them in more detail below to allow them to give further consideration to them. 1) The point about non-English studies has been side-stepped. The studies are referenced and their contents alluded to in James, D.V., Farnham, F. R., Sukhwal, S., Jones, K., Carlisle, J., & Henley, S. (2015). Aggressive/ intrusive behaviours, harassment and stalking of Westminster members of parliament: A prevalence survey and cross-national comparison, Journal of Forensic Psychiatry and Psychology, doi: 10.1080/14789949.2015.1124908 It is not necessary to understand Dutch, Swedish or Norwegian to access the contents of the originals. 2) Some reference to the reason why a higher proportion of MPs should experience poor mental health might be given. The above paper on MPs finds that a higher proportion of MPs than the general public experience stalking and harassment. The second paper (referenced in the manuscript) details the symptoms that MPs experience in consequence. Given the overlap between these and items in the GHQ and the compelling research evidence that stalking results in higher levels of mental disorder, I would have thought that some comment about the possible relevance of this would have been included in the discussion. This is, after all, a highly topical issue at present.
--

VERSION 2 – AUTHOR RESPONSE

Reviewers' Comments to Author:

Reviewer: 3

The authors have not responded as I would have hoped to two of my initial comments. I would recommend publication as is, should the authors not wish at this stage to address them. However, I am repeating them in more detail below to allow them to give further consideration to them.

The point about non-English studies has been side-stepped. The studies are referenced, and their contents alluded to in James, D.V., Farnham, F. R., Sukhwal, S., Jones, K., Carlisle, J., & Henley, S. (2015). Aggressive/ intrusive behaviours, harassment and stalking of Westminster members of parliament: A prevalence survey and cross-national comparison, *Journal of Forensic Psychiatry and Psychology*, doi: 10.1080/14789949.2015.1124908 It is not necessary to understand Dutch, Swedish or Norwegian to access the contents of the originals.

Some reference to the reason why a higher proportion of MPs should experience poor mental health might be given. The above paper on MPs finds that a higher proportion of MPs than the general public experience stalking and harassment. The second paper (referenced in the manuscript) details the symptoms that MPs experience in consequence. Given the overlap between these and items in the GHQ and the compelling research evidence that stalking results in higher levels of mental disorder, I would have thought that some comment about the possible relevance of this would have been included in the discussion. This is, after all, a highly topical issue at present.

□ Thank you for your comment and for providing this reference. We agree that stalking, harassment and intrusive behaviours are relevant and we have added details in the discussion (lines 315f, and 356-361).

□ We did reference and point out the cross-national comparison for these stalking/harassment studies. These studies focus on stalking and harassment, with a strong focus on the mental health of the perpetrators. Our study is a first study on understanding mental health of MPs, rather than exploring the behaviours that evoke mental health problems. We have therefore decided against elaborating further into the other international studies on stalking and harassment. We did however include bullying, harassment and stalking in the implications for future research, as needed prevention and better understanding of the causes for mental health problems and specific risk factors in MPs. (line 405)